# OpenReview forum: "AdaNav: Adaptive Reasoning with Uncertainty for Vision-Language Navigation"
_ICLR.cc/2026/Conference — Submitted to ICLR 2026_

### Official Review · Reviewer_a5ZA · 2025-10-20

**Soundness:** 4
**Presentation:** 2
**Contribution:** 4
**Rating:** 8
**Confidence:** 3

**Summary:**

The authors introduce AdaNav, a module that allows VLN agents to select a reasoning mode at a given step. The module takes as input the history of action entropies and current state. The output is a probability to invoke one of four reasoning modes: summary, error correction, description and "no reasoning". In contrast to previous work, this setup focuses reasoning to critical points in the trajectory in order to avoid context rot. The reasoning mode policy is initialized by a heuristic based on action entropy and slowly annealed towards a fully learned policy. The reward for the reasoning mode policy takes success rate, distance-to-goal and reasoning output length into account. The reasoning module can be dropped into existing agents and experiments show consistent improvements across benchmarks. Ablation experiments show that the approach is robust to hyperparameter and design choices like heuristic prior are crucial for training.

**Strengths:**

- well motivated approach
- important problem of selective reasoning in VLN
- good ablations

**Weaknesses:**

- presentation is a bit cramped but could be solved by additional page

**Questions:**

Did you consider outdoor VLN tasks? If not, why?

---

> ### Author Response · Authors · 2025-11-19
>
> **We sincerely appreciate your positive feedback on our work, and we welcome any questions or further discussion.**
>
> ---
>
> ## **Weakness 1 :Presentation is a bit cramped**
>
> Thank you for your reminder. We have revised the paper's formatting and hope this resolves your concern.
>
> ---
>
> ## **Question1: Challenges and Preliminary Exploration in Outdoor VLN Tasks**
> > **Summary:** Current VLN base models perform poorly on outdoor tasks, but AdaNav improves performance by more effectively leveraging VLM pretraining knowledge through adaptive reasoning, highlighting its potential for indoor and outdoor VLN  research. We have included these comparisons in the revised manuscript, **in Appendix G and line 412**.
>
> This is an important question. Due to the difficulty of **data collection for embodied VLN**, existing open-source navigation models face three major limitations:
>
> 1. **Training data scarcity for outdoor VLN:** Most available datasets are collected from **indoor VLN tasks**, leaving outdoor navigation largely unexplored.
> 2. **Limited generalization from indoor to outdoor:** Even indoor VLN data is insufficient for models to generalize effectively to outdoor tasks.
>
> As a result, current VLN models perform poorly on outdoor VLN benchmarks, as shown below.
>
> |  | OPENUAV | OpenUAV | OpenUAV | OpenUAV | Real World-Simple |Real World-Simple |Real World-Complex |Real World-Complex |
> |:---:|:---:|:---:|:---:|:---:|:---:|:---:|:---:|:---:|
> |  | NE ↓| SR↑ | OSR↑ | SPL↑ | SR↑ |NE↓ |SR↑ |NE↓ |
> | CMA | 102.92 | 14.83 | 22.49 | 13.90 | - |- |- |- |
> | Navila | 93 | 21.5 | 45.1 | 21.5 | 55.0 |2.5|35.0|3.5|
> | Navila-AdaNav | 85 | 25.5 | 50.1 | 23.8 | 67.0 |1.9|45.0|2.5|
>
> To explore this direction, we conducted experiments on the **OpenUAV outdoor VLN benchmark** as well as in **real-world environments**. Even **without access to any outdoor training data**, **AdaNav** was able to **enhance the performance of the base model**. We attribute this improvement to AdaNav's reasoning mechanism, which leverages knowledge acquired by the base VLM from **large-scale internet datasets during pretraining**, partially compensating for the scarcity of outdoor data during post-training.
>
> Nevertheless, **outdoor VLN remains an underexplored area** in the community, and AdaNav provides a **promising direction and reference** for future research in this domain.
>
> We hope that our response addresses your concerns.

---

> ### Author Response · Authors · 2025-11-28
>
> Dear Reviewer,
>
> I hope this message finds you well.
>
> We would like to **express our sincere gratitude** for your thoughtful review and would like to **gently follow up** on our rebuttal submitted earlier. We understand that you may be busy, and we truly appreciate the time and effort you have already dedicated to reviewing our work.
>
> **If there are any additional questions or points that would benefit from further clarification, we would be more than happy to provide more details or continue the discussion**.
>
> **Thank you again** for your constructive and supportive feedback.
>
> Best regards,
> Authors

---

### Official Review · Reviewer_BZM3 · 2025-10-30

**Soundness:** 2
**Presentation:** 3
**Contribution:** 2
**Rating:** 4
**Confidence:** 3

**Summary:**

This paper introduces an uncertainty-driven framework that decides when and how to reason during VLN via a lightweight Uncertainty-Adaptive Reasoning (UAR) block and a Heuristic-to-RL training scheme. Action Entropy serves as an interpretable trigger signal, then reinforcement learning refines the policy.

**Strengths:**

1. Clear formulation that separates navigation and reasoning policies and factors the reasoning decision into “when/which mode” and “what to say,” which makes the mechanism easy to slot into existing VLN stacks.
2. Sensible uncertainty signal backed by a diagnostic showing higher action entropy in failed trajectories, plus a compact control vector for mode selection.
3. Practical Heuristic-to-RL schedule that starts from uncertainty priors and anneals toward a learned policy, avoiding costly reasoning supervision.

**Weaknesses:**

1. The approach hinges on action entropy. Alternatives such as value uncertainty, disagreement, or temporal variance are not compared head-to-head, even though the paper shows a correlation analysis for entropy.

2. Reward shaping penalizes reasoning by length. This may incentivize terse but unhelpful traces or gaming of minimal length; failure-case analysis for this trade-off is limited.

3. Additional backbones or continuous-control tasks would make the “plug-in” claim feel more comprehensive.

**Questions:**

1.What is the end-to-end latency budget on hardware during real-world runs and how much of it is attributable to UAR vs. text reasoning vs. base policy inference. A simple table with mean/95th percentile would help practitioners.

2. When the entropy prior disagrees with the learned selector during annealing, does the system exhibit oscillation or premature triggering. Any guardrails beyond λ-scheduling.

---

> ### Author Response · Authors · 2025-11-19
>
> We **sincerely appreciate the reviewer’s valuable feedback**.
>
> ---
>
> ## **Weakness1: Comparison of Alternative Uncertainty Measures**
>
> > **Summary:** Action entropy proves to be the most effective uncertainty signal (offering higher navigation performance, and better reasoning efficiency than value uncertainty, value disagreement, or temporal variance). We have included these comparisons in the revised manuscript, **in Appendix F**.
>
> We would like to reiterate our motivation and key contribution:
>
> - This work, to the best of our knowledge, is the first navigation framework that leverages **uncertainty signals** to guide **adaptive reasoning**
>
> - **In scenarios where embodied interaction data is extremely costly**, AdaNav can learn difficulty-aware reasoning policies from **just 6K simulation samples** via adaptive reasoning, significantly improving navigation performance. Impressively, it allows **open-source models to outperform closed-source counterparts trained on million-scale data**, demonstrating both **data efficiency** and **practical scalability**.
>
>
> As demonstrated in Section 3, **action entropy** provides a strong indicator of the model’s confidence and the difficulty of the current situation, which is why we adopt it as our primary uncertainty signal. Following the reviewer’s suggestion, we further designed and evaluated several alternative uncertainty measures: **value uncertainty**, **value disagreement**, and **temporal variance**. All these signals were trained using the same **Heuristic-to-RL training pipeline** with 6K samples.
>
> ---
>
> ### • Value Uncertainty
>
> Following common practice in prior work [1], we trained an additional **critic head** for policy to estimate the expected return.
> The variance across multiple value heads
>
> $
> \sigma_V^2 = \text{Var}(V_t^{(i)})
> $
>
> serves as a measure of **value-based uncertainty**, capturing the inconsistency in future reward estimation.
>
> ---
>
> ### • Value Disagreement
> **Value disagreement** quantifies the diversity among predictions of multiple critic heads (the 6K training samples are split into three subsets, and one critic head is trained on each subset).
> It can be computed as the **average pairwise squared difference** between critic heads:
>
> $D_V = \frac{2}{N(N-1)} \sum_{i<j} (V_t^{(i)} - V_t^{(j)})^2$
>
> where $V_t^{(i)}$ denotes the value predicted by the *i-th* head at time *t*.
> Larger $D_V$ indicates greater disagreement among the value heads.
>
> ---
>
> ### • Temporal Variance
>
> **Temporal variance** measures short-term fluctuations of action entropy over a sliding window of length $w$:
>
> $
> \sigma_H^2 = \frac{1}{w} \sum_{k=t-w+1}^{t} (H(\pi_k) - \bar{H})^2, \quad
> \bar{H} = \frac{1}{w} \sum_{k=t-w+1}^{t} H(\pi_k)
> $
>
> where $H(\pi_k)$ is the **entropy of the action distribution** at step *k*.
> Large $\sigma_H^2$ indicates **unstable decision confidence**.
>
> | Component     | NE↓ | OS↑ | SR↑ | SPL↑ |Avg.reasoning_num|
> |:---:|:---:|:---:|:---:|:---:|:---:|
> | Navid | 5.47 | 49.0 | 37.0 |  35.0| 0 |
> | Value Uncertainty | 5.41 | 54.31 | 44.69 | 40.67 | 4.8|
> | Value Disagreement| 5.44|  54.78| 43.73   | 40.53 | 5.5|
> | Temporal Variance | 5.40 | 54.65 | 45.21 | 41.15 | 4.2|
> | Navid-AdaNav | 5.39 | 57.89 | 47.7 | 42.34 | 2.5 |
>
> As shown in the table, we attribute the superior performance to **action entropy $H_{\le t}$**, which more accurately captures the model’s **intrinsic uncertainty** and provides a **better perception of task difficulty**, leading to **higher navigation performance** and **greater efficiency** compared with alternative uncertainty measures.
>
> ## **Weakness2: Clarification on Reward Shaping Design and Its Effect on Reasoning Length**
>
> > **Summary:** Using the minimal successful length as the standard ensures no incentive to over-shorten, maintaining a clear balance between efficiency and effectiveness.
>
> You are absolutely correct that the design of reward shaping is crucial. However, please note that in **Equation 11**, the *standard group* is defined as the **minimal generation length among successful samples**, which ensures that—**conditioned on success**—there is no incentive to deliberately shorten the reasoning length to take shortcuts.
>
> This design establishes a **balanced trade-off between efficiency and effectiveness**, encouraging concise yet meaningful reasoning rather than superficial brevity.

---

> > ### Author Response · Authors · 2025-11-19
> >
> > ## **Weakness3: Additional Experiments with Newly Released Uni-NaVid Backbone**
> >
> > >**Summary:** Using the newly released Uni-NaVid backbone, along with evaluations on additional continuous-control tasks, further confirms that AdaNav consistently improves navigation performance across diverse models and environments. We have included these section in the revised manuscript, **in Appendix E**.
> >
> > Thank you for your valuable suggestion. As the availability of open-source navigation models is quite limited, we initially selected **all the state-of-the-art open-source backbones** available at the time of submission — namely, **NaVid** and **NaviLA** — for our experiments.
> >
> > Fortunately, about one month after our paper submission, **Uni-NaVid** was released as an open-source model. Following your suggestion, we conducted **additional experiments using Uni-NaVid as the backbone**, and further evaluated our framework on **more continuous-control tasks**, including:
> >
> > - Object Goal Navigation on Matterport3D, and
> > - Object Goal Navigation on HM3D-OVON.
> >
> > The experimental results are shown below:
> >
> > | Method | SR | SPL |
> > |:---:|:---:|:---:|
> > | PIRLNav-IL-RL | 70.4 | 34.1 |
> > | OVRL | 62.0 | 26.8 |
> > | OVRL-v2 | 64.7 | 28.1 |
> > | Uni-NaVid | 73.7 | 37.1 |
> > | Uni-NaVid-AdaNav | 80.15 | 42.2 |
> >
> > | Method | VAL SEEN | VAL SEEN | VAL SEEN SYNONYMS | VAL SEEN SYNONYMS | VAL UNSEEN | VAL UNSEEN |
> > |:---:|:---:|:---:|:---:|:---:|:---:|:---:|
> > | - | SR | SPL | SR | SPL | SR | SPL |
> > | DAgRL | 41.3 | 21.2 | 29.4 | 14.4 | 18.3 | 7.9 |
> > | VLFM | 35.2 | 18.6 | 32.4 | 17.3 | 35.2 | 19.6 |
> > | DAgRL+OD | 38.5 | 21.1 | 39.0 | 21.4 | 37.1 | 19.8 |
> > | Uni-NaVid | 41.3 | 21.1 | 43.9 | 21.8 | 39.5 | 19.8 |
> > | Uni-NaVid-AdaNav | 50.62 | 25.62 |  52.61| 27.8 | 47.8 | 26.2 |
> >
> > | Method | R2R | R2R | R2R | R2R | R2R | RxR | RxR | RxR | RxR | RxR |
> > |:---:|:---:|:---:|:---:|:---:|:---:|:---:|---|---|---|---|
> > | - | TL | NE↓ | OS↑ | SR↑ | SPL↑ | TL | NE↓ | OS↑ | SR↑ | SPL↑ |
> > | Uni-NaVid | 9.71 | 5.58 | 53.3 | 47.0 | 42.7 | 15.8 | 6.24 | 55.5 | 48.7 | 40.9 |
> > | Uni-NaVid-AdaNav | 9.91 | 5.11 | 61.6 | 56.9 | 49.5 | 17.7 | 6.2 | 59.6 | 54.5 | 44.2 |
> >
> >
> > ---
> >
> > ## **Question1:Clarification on Computational Efficiency of the UAR Block**
> > > **Summary**: The UAR Block adds negligible latency and, by adaptively limiting reasoning to about 2.5 steps per trajectory, maintains nearly unchanged overall task time while substantially improving navigation performance. We have included these details in the revised manuscript, **which can be found in the ablation study section**.
> >
> > We sincerely thank the reviewer for this important suggestion and provide the following comparison to clarify computational efficiency.
> >
> > | Component | Peak.Latency | Avg.Latency | P90.Latency|
> > |:---:|:---:|:---:|---|
> > | UAR |0.0039s|0.0029s|    0.0037s  |
> > | Base Action Policy | 1.31s | 0.76s |  1.00s   |
> > | Text Reasoning | 5.01s | 2.94s  |   3.81s   |
> >
> > > *P90.Latency represents the time value below which 90% of the sample measurements fall.*
> >
> > As shown in the table above, the **UAR Block introduces negligible latency** — its additional computation accounts for **less than 0.4% of backbone inference time**.
> >
> > Although a single text reasoning step is approximately **3.9 times slower** than the base policy inference, the **adaptive reasoning mechanism** learned by the UAR Block reduces the number of reasoning steps per trajectory to only **2.5**, over trajectories with an average length of **55 steps**.
> >
> > Consequently, the **overall end-to-end task completion time remains nearly unchanged**, while **navigation performance is substantially improved**.
> >
> >
> >
> > ## **Question2: Stabilization of the Annealing Stage**
> >
> > We thank the reviewer for raising this insightful question.
> >
> > First, the λ-scheduling mechanism effectively regulates the interplay between the entropy prior and the learned policy, and we **did not** observe any noticeable oscillation during training.
> >
> > Moreover, thanks to the **inherent properties of RL**, the policy can learn from trajectories **regardless of whether they are successful or failed**, using all experiences to update and improve performance, in contrast to SFT-based methods that rely solely on supervised examples. This prevents overfitting to a narrow demonstration distribution and enables the policy to converge toward a more robust and better-generalizing solution. Therefore, even if minor oscillations were to occur, they **would not** destabilize training or hinder convergence.
> >
> >
> >
> > [1] Wang X, Huang Q, Celikyilmaz A, et al. Reinforced cross-modal matching and self-supervised imitation learning for vision-language navigation[C]//Proceedings of the IEEE/CVF conference on computer vision and pattern recognition. 2019: 6629-6638.

---

> ### Author Response · Authors · 2025-11-28
>
> Dear Reviewer,
>
> I hope this message finds you well.
>
> We would like to **express our sincere gratitude** for your thoughtful review and would like to **gently follow up** on our rebuttal submitted earlier. We understand that you may be busy, and we truly appreciate the time and effort you have already dedicated to reviewing our work.
>
> **If there are any additional questions or points that would benefit from further clarification, we would be more than happy to provide more details or continue the discussion**.
>
> **Thank you again** for your constructive and supportive feedback.
>
> Best regards,
> Authors

---

### Official Review · Reviewer_7XEJ · 2025-10-31

**Soundness:** 3
**Presentation:** 2
**Contribution:** 3
**Rating:** 4
**Confidence:** 4

**Summary:**

This paper propose AdaNav, an adaptive navigation inference framework based on uncertainty. Combining prior and heuristic reinforcement learning training mechanisms, it significantly outperforms multiple closed-source models in benchmarks such as R2R and RxR-CE, as well as in real-world scenarios, without relying on large-scale labeled data.

**Strengths:**

1.An adaptive inference framework is proposed that does not require large-scale inference labeled data, addressing the key questions of "when to infer and how to infer" in VLN.

2.Innovatively, action entropy is used as an uncertainty signal, combined with historical trends, to dynamically trigger inference.

3.This paper proposed a "Heuristic-to-RL" training mechanism, combining heuristic guidance and reinforcement learning for optimization.

**Weaknesses:**

1. The figures need improvement. Figure 1 mainly shows the differences from traditional methods, but the description of the prior and UAR Block is very limited. Specific figures or sub-figures describing the method could be added.

2. Regarding the effectiveness of the UAR Block and the necessity of reinforcement learning, there is a lack of comparison with adaptive inference baselines, such as inference based on fixed interval changes or triggering based on the simplest confidence level.

3. I am curious about the computational efficiency metrics, such as the average inference time and memory usage of this adaptive module. It is difficult to assess its feasibility in practical deployment.

4. I am curious about the selection of the weights $W_1$, $W_2$, and $W_3$ in Equation 6, as well as b. Is this learnable? If only action entropy has a significant impact? An ablation study could be conducted.

**Questions:**

1. In Figure 3, the action entropies for success and failure overlap considerably, especially at the beginning and end of the episode. I'm confused about the definition of action entropy as a good measure of uncertainty in the paper. How is this difficult-to-determine uncertainty handled?

2. Figure 4b shows a relatively high frequency of actual step-interval inference. Is adaptive uncertain inference necessary? Table 7(a) uses a fixed 5-step inference; is it possible that adaptive uncertain inference has a higher actual inference frequency? What if the ablation is a fixed 3-step, 1-step inference?

3. In Heuristic-to-RL training, how sensitive is the prior distribution setting? Are progress- and distance-based prior strategies better than action entropy?

4. Table 3 uses significantly different data for 8-frame and 64-frame models, but the model results are not significantly different. Is this because the model has a limited learnable context length?

5. For failure case analysis, under what circumstances did the UAR Block fail to trigger inference correctly?

---

> ### Author Response · Authors · 2025-11-19
>
> We thank the reviewer for the thoughtful feedback and appreciate the recognition of our contributions, including the **adaptive reasoning framework**, the use of **action entropy**, and the **Heuristic-to-RL training mechanism**. We would like to reiterate our motivation and key contribution:
>
> - This work, to the best of our knowledge, is the first navigation framework that leverages **uncertainty signals** to guide **adaptive reasoning**
>
> - **In scenarios where embodied interaction data is extremely costly**, AdaNav can learn difficulty-aware reasoning policies from **just 6K simulation samples** via adaptive reasoning, significantly improving navigation performance. Impressively, it allows **open-source models to outperform closed-source counterparts trained on million-scale data**, demonstrating both **data efficiency** and **practical scalability**.
>
> - In terms of efficiency, AdaNav averages only **2.5 reasoning steps per 55-step trajectory** (roughly **once every 20 steps**), achieving a **significant efficiency improvement**
>
> ---

---

> ### Author Response · Authors · 2025-11-19
>
> ## **Weakness1: The figures need improvement.**
>
> We appreciate your feedback. Figure 1 has been enhanced to better illustrate the prior. Specifically, we visually depict the **entropy accumulation process to obtain H_score** and the **entropy-to-prior mapping**, and we also show the **UAR Block structure**, which consists of an MLP followed by a Softmax. Please see the revised version.
>
> ## **Weakness2: Explanation of the Effectiveness of the UAR Block and the Necessity of RL**
> > **Summary**: Navid-AdaNav consistently outperforms all non-RL baselines, i.e., fixed-interval or heuristic confidence-level baselines,by 20% SR..
>
> We appreciate the reviewer’s insightful comment regarding the comparison with adaptive inference baselines.
> To verify whether reinforcement learning and the proposed UAR Block offer additional benefits over simpler heuristics, we implemented three representative baselines in Table 7a:
>
> 1. **Effectiveness of the UAR**: fixed-interval inference triggering (5 steps, labeled “w/o UAR*”) and random triggering (“w/o UAR†”);
>
> 2. **Effectiveness of RL**: inference triggered only by a fixed confidence threshold without RL optimization (“w/o RL”).
>
> For your convenience, we have copied it below.
> |Component|NE↓|OS↑|SR↑|SPL↑|
> |:---:|:---:|:---:|:---:|:---:|
> |Navid |5.47|49.0|37.0|35.0|
> |w/o UAR*|5.45|53.25|40.12|38.83|
> |w/o UAR†|5.44|52.10| 39.5|38.65|
> |w/o RL| 5.44|55.33|42.53|39.65|
> |Navid-AdaNav|5.39|57.89|47.7|42.34|
>
> It can be observed that, compared with the baselines, incorporating the **RL-based UAR Block** yields an improvement of approximately **20%**.
>
> Furthermore, to provide more comprehensive evidence, **we additionally evaluated a series of fixed-interval (1, 3, 7, 9, 15)**  reasoning strategies without RL training.
>
> |Component| NE↓ | OS↑ | SR↑ | SPL↑ |
> |:---:|:---:|:---:|:---:|:---:|
> |Navid | 5.47|49.0|37.0|35.0|
> |fixed 1-step|5.49|51.2|38.2|36.5|
> |fixed 3-step|5.47|50.35|39.51|37.87|
> |fixed 5-step|5.45|53.25|40.12|38.83|
> |fixed 7-step|5.45|52.56|40.1| 37.55|
> |fixed 9-step|5.43|51.25|40.02| 38.35|
> |fixed 15-step|5.45|52.23|39.2| 37.48|
> |Navid-AdaNav|5.39|57.89|47.7|42.34|
>
> > Note: AdaNav averages only 2.5 reasoning steps per 55-step trajectory(**roughly once every 20 steps**) far less frequent than the above fixed-step baseline.
>
> We have included these details in the revised manuscript, **in Appendix I**.
>
> ---
>
> ## **Weakness3: Explanation of Computational Efficiency**
> > **Summary**: The UAR Block adds negligible computational overhead. We have included these details in the revised manuscript, **which can be found in the ablation study section**.
>
> The computational efficiency of the UAR Block is indeed a critical consideration.
> As described in Section 3.2.1, **it is a lightweight module** with less than **0.4%** of the backbone’s parameters.
> The following table reports the average inference time and memory usage for generating a single action, showing that **the additional overhead compared to the backbone is negligible**.
> |  | Inference Time  | Memory Usage |
> |:---:|:---:|:---:|
> |Navid backbone|0.76 s|14 GB|
> |UAR Block|0.0029 s|33.6 MB|
>
> ## **Weakness4: Explanation of Parameter Learning and the Role of Action Entropy in Eq. (6)**
> > **Summary:** Combining action entropy with contextual observation features allows AdaNav to achieve both efficient reasoning and superior navigation performance, avoiding the trade-offs seen when using either signal alone. We have included these details in the revised manuscript, **which can be found in the ablation study section**.
>
> 1. All parameters in Eq. (6) are trainable and optimized jointly with the RL objective.
>
> 2. As demonstrated in Section 3.2.1, **action entropy** serves as a reliable statistical indicator of model uncertainty, providing an interpretable and stable heuristic prior for RL training.
>
>     Meanwhile, contextual features from observations and instructions remain essential, as they enable the policy to **ground uncertainty in the physical environment and linguistic intent**.
>
>     Through **reinforcement learning**, AdaNav **progressively aligns** three key elements:**interaction dynamics** (environmental observations), **internal uncertainty** (action entropy), and **task intent** (user instructions), enabling the agent to develop a coherent and difficulty-aware reasoning policy.
>
> |Component|NE↓|OS↑|SR↑|SPL↑|Avg.infer_num|
> |:---:|:---:|:---:|:---:|:---:|:---:|
> |Navid|5.47|49.0|37.0|35.0|0|
> |Only Action Entropy|5.41|56.73|45.23|41.55|4.5|
> |Only Observation|5.43|55.23|43.12|40.74|8.7|
> |Navid-AdaNav|5.39|57.89|47.7|42.34|2.5|
>
> As shown in the table above, when trained with the same 6K samples, models that use **only action entropy** or **only observation features** exhibit a **2–4× increase** in reasoning frequency but suffer a **5–10% drop** in navigation success rate.
> This result highlights that **combining contextual features with uncertainty signals is essential** for achieving both **efficiency** and **high-performance navigation**.

---

> ### Author Response · Authors · 2025-11-19
>
> ## **Question1: On the Reliability of Action Entropy and the Handling of Ambiguous Uncertainty**
> > **Summary:** Action entropy provides a stable heuristic prior during RL cold-start, and through joint optimization with observations and task instructions, AdaNav learns a coherent, difficulty-aware reasoning policy without relying on fixed entropy thresholds.
>
> We believe this concern relates to  Weakness 4.
> In our framework, action entropy is used purely **as a statistical indicator of action uncertainty**. It provides an interpretable and stable heuristic prior that is especially beneficial **during the RL cold-start stage**, where training data is limited.
>
> As illustrated in Figure 3, this entropy prior delivers a **strong, reliable, and easy-to-estimate** uncertainty signal **across a wide range of scenarios**.
>
> During subsequent RL training, the policy gradually learns to **align environment interaction dynamics (observations), internal uncertainty estimates (action entropy), and task-specific instructions**. This process equips the UAR Block with greater generalization ability and adaptivity.
>
> Through this joint optimization, the agent ultimately learns a coherent and difficulty-aware reasoning policy—one leverages entropy as a useful cue **without being constrained by fixed entropy thresholds**.
>
> ## **Question2: Regarding the Necessity of Adaptive Uncertain Reasoning**
>
> > **Summary:** AdaNav demonstrates that **adaptive uncertain reasoning**(triggered selectively rather than excessively) is essential for balancing **efficiency** and **effectiveness**. It enables timely reasoning that improves navigation performance without incurring unnecessary computational cost or overthinking.
>
> As shown in **Figure 4a**, AdaNav averages only **2.5 reasoning steps per 55-step trajectory**, roughly **once every 20 steps**, far less frequent than the fixed 5-step baseline.
>
> We demonstrate the necessity of Adaptive Reasoning from **both efficiency and effectiveness perspectives**, highlighting AdaNav’s strong performance in these aspects.
>
> * **Efficiency**: We measured end-to-end latency for UAR, text reasoning, and base policy action inference over 100 runs as follow. A single reasoning step incurs approximately 3.9 times the latency of a base action inference. Consequently, **excessive reasoning can significantly degrade real-world deployment efficiency**.
>
> | Component | Peak.Latency | Avg.Latency | P90.Latency|
> |:---:|:---:|:---:|---|
> | UAR |0.0039 s|0.0029 s|    0.0037 s  |
> | Base Action Policy | 1.31 s | 0.76 s |  1.00 s   |
> | Text Reasoning | 5.01s | 2.94s  |   3.81s   |
>
> > *P90.Latency represents the time value below which 90% of the sample measurements fall.*
>
> * **Effectiveness**: Excessive reasoning can even lead to an **“overthinking”**[1,2,3] effect, where frequent reasoning triggers increase computational cost without improving success rate. This is further supported by our fixed step-interval experiments (1, 3, 7, 9, 15 steps) discussed in Weakness 2, which show that **higher reasoning frequency does not necessarily yield better performance**.
>
> **AdaNav Performance**: **Figure 4b** shows the proportion of each reasoning mode(**measured every 5 steps**), rather than the actual reasoning frequency, which is captured in **Figure 4a**. As shown in Fig.4a, **the overall reasoning frequency per navigation episode is very low**: on average only **2.5 times over ~55-step trajectories**, indicating that the model effectively avoids over-triggering reasoning.
>
> Moreover, combining **Figure 4b and Table 6**, AdaNav reasons **only when necessary**, reducing overall reasoning frequency while ensuring **timely and accurate decisions**, which improves navigation performance.
> Together, these findings demonstrate that reasoning at the right moments, rather than more frequently, is crucial, **underscoring the necessity and advantage of adaptive uncertain reasoning**.

---

> > ### Author Response · Authors · 2025-11-19
> >
> > ## **Question3: Robustness to Prior Distribution and Superiority of Action Entropy**
> > > **Summary:** AdaNav is robust to prior distributions, and action entropy provides a more accurate and general measure of the model’s internal confidence than task progress or distance.
> >
> > 1. We evaluated AdaNav's sensitivity to 15 different prior distributions in Table 7(b) and 7(c). The results remain stable (SR：47.7～49.6，**variation <3%**), showing that **AdaNav is robust to prior distribution**.
> >
> > 2. In navigation tasks, progress or distance cannot reliably indicate whether the current behavior is good or whether reasoning is necessary, since **temporarily moving away from the goal may be part of a better long-term plan to reach the destination**.
> >
> > Below are the comparative results under the **distance-based prior** setting:
> > | Component     | NE↓ | OS↑ | SR↑ | SPL↑ |Avg.reasoning_num|
> > |:---:|:---:|:---:|:---:|:---:|:---:|
> > | Navid | 5.47 | 49.0 | 37.0 |  35.0| 0 |
> > | Distance-based prior | 5.43 | 54.13 | 45.75 | 40.89 | 7.2|
> > | Action Entropy prior(AdaNav) | 5.39 | 57.89 | 47.7 | 42.34 | 2.5 |
> >
> > As shown above, the distance-based prior triggers **nearly 3× more reasoning steps**, yet offers weaker performance gains.
> > In contrast, **only the model’s intrinsic action entropy captures its true internal confidence**, providing a **more accurate and efficient uncertainty signal** that is not tied to trajectory geometry or heuristic distance metrics.
> >
> > ## **Question4: Context-Length Differences Reflect Backbone Capacity, Not AdaNav Limitations**
> >
> > > **Summary:** Context-length differences reflect the backbone’s capacity, and AdaNav—via the lightweight UAR Block—complements rather than limits the base model’s perception abilities.
> >
> > We would like to clarify that **AdaNav does not modify the backbone architecture**; it only introduces the lightweight UAR Block as a plug-in module. Therefore, variations in context length primarily reflect the base model’s capacity **rather than any limitation of AdaNav**.
> >
> > As shown in Table 3, incorporating **AdaNav does not compromise the backbone’s spatial scene understanding**. On the contrary, it leads to slight improvements, demonstrating that our adaptive reasoning mechanism **complements rather than interferes with the base model’s perception capabilities**.
> >
> >
> > ## **Question5: Analysis of UAR Block Failure Cases and Improvement**
> > > **Summary:** The UAR Block may miss reasoning under transient ambiguities; however, targeted RL fine-tuning on specific scenarios significantly reduces such failures. We have included these comparisons in the revised manuscript, **in Appendix H**.
> >
> > The UAR Block primarily fails to trigger reasoning in cases of transient ambiguity, such as vague instructions or sudden changes in lighting and viewpoint. In these situations, action entropy fluctuates briefly but does not remain high enough to surpass the reasoning threshold, causing the system to skip reasoning when it would be beneficial.
> >
> > Fortunately, **the frequency of such failures decreases significantly after RL fine-tuning on data specifically designed to cover these scenarios**. In this process, the model learns to better associate transient uncertainty with the need for reasoning. **To validate this, we collected 500 samples exhibiting instruction vagueness and lighting changes and conducted additional evaluations on R2R and RxR**. The results indicate that tailoring the training environment to address these specific challenges can further improve the model’s navigation performance.
> >
> >
> > | Method | R2R | R2R | R2R | R2R | RxR | RxR | RxR | RxR |
> > |:---:|:---:|:---:|:---:|:---:|:---:|:---:|:---:|:---:|
> > | -  | NE↓ | OS↑ | SR↑ | SPL↑  | NE↓ | OS↑ | SR↑ | nDTW↑ |
> > | Navid  | 5.47 |49.0 |37.0 |35.0| 6.79| 46.2 |40.5 |52.2|
> > | NaVid-AdaNav |  5.39| 57.89 |47.7 |42.34| 6.38| 58.1 |47.01 |56.8|
> > | NaVid-AdaNav* | 5.35 | 58.72 | 48.85 | 42.88 | 6.12 | 59.62 | 48.55 | 57.2|
> >
> > > **Note:** *NaVid-AdaNav\** refers to the model after additional training with 500 samples covering instruction vagueness and lighting changes.
> >
> > [1] Sui Y, Chuang Y N, Wang G, et al. Stop overthinking: A survey on efficient reasoning for large language models[J]. arXiv preprint arXiv:2503.16419, 2025.
> >
> > [2]Wu Y, Wang Y, Ye Z, et al. When more is less: Understanding chain-of-thought length in llms[J]. arXiv preprint arXiv:2502.07266, 2025.
> >
> > [3] Shen Y, Zhang J, Huang J, et al. Dast: Difficulty-adaptive slow-thinking for large reasoning models[J]. arXiv preprint arXiv:2503.04472, 2025.

---

> ### Author Response · Authors · 2025-11-28
>
> Dear Reviewer,
>
> I hope this message finds you well.
>
> We would like to **express our sincere gratitude** for your thoughtful review and would like to **gently follow up** on our rebuttal submitted earlier. We understand that you may be busy, and we truly appreciate the time and effort you have already dedicated to reviewing our work.
>
> **If there are any additional questions or points that would benefit from further clarification, we would be more than happy to provide more details or continue the discussion**.
>
> **Thank you again** for your constructive and supportive feedback.
>
> Best regards,
> Authors

---

### Meta-Review · Area_Chair_Dyo7 · 2025-12-30

**Summary:**

I would like to acknowledge the thorough engagement of the authors with the reviewers’ comments and their substantial efforts to provide more experiments and clarifications. At this time, I suggest rejecting this paper, informed by reviewer feedback as well as my own reading of the manuscript. At the end of this summary, I provide a set of highly relevant works in vision-language navigation that have been developed by the robotics community over the past ~4 years that I recommend discussing and potentially benchmarking against to strengthen the manuscript for a future submission.

**7XEJ** states that the *strengths* are the method’s ability to operate without large-scale inference labeled data, how action entropy is used as uncertainty signal, and the "Heuristic-to-RL" training approach. The *concerns* come from lack of clarity in the figures, lack of comparison to adaptive inference baselines, need for ablations to study the sensitivity of the method, and some detailed questions about the results and methodology.

**BZM3** states that the *strengths* are a design which separates navigation and reasoning policies, a practical heuristic-to-RL pipeline, and a reasonable measure of uncertainty in the actions. The main *concerns* come from how alternative forms of uncertainty quantification beyond action entropy weren’t compared head-to-head despite the uncertainty being a key aspect of the method, the heuristic reward-shaping, and not fully demonstrating the “plug and play” nature of the method because there aren’t comparisons to other backbones.

**a5ZA** states that the *strengths* are the well-motivated approach, important problem of selective reasoning, and good ablations. The main *concerns* are the paper’s presentation and if outdoor scenarios were considered in the evaluation.

From my own reading, I believe there are some key missing prior works in VLN and uncertainty-aware VLN that should be contextualized.
* [1] Ren, Allen Z., et al. "Explore until confident: Efficient exploration for embodied question answering." Robotics: Science and Systems (2024).
* [2] Ke, Liyiming, et al. "Tactical rewind: Self-correction via backtracking in vision-and-language navigation." Proceedings of the IEEE/CVF conference on computer vision and pattern recognition. 2019.
* [3] Hirose, Noriaki, et al. "Lelan: Learning a language-conditioned navigation policy from in-the-wild videos." Conference on Robot Learning (2024).
* [4] Shah, Dhruv, Błażej Osiński, and Sergey Levine. "Lm-nav: Robotic navigation with large pre-trained models of language, vision, and action." Conference on robot learning. PMLR, 2023.

I recommend the authors connect more with recent development in robotics, which has been advancing vision-language navigation over the past several years. For example, I recommend investigating paper [1] in more detail, since it is the closest paradigm of uncertainty-aware vision-language navigation. Instead of maintaining uncertainty over lower-level actions, it maintains uncertainty over the high-level visual Q&A tasks that then inform what low-level actions to take. This would be a helpful to benchmark or discuss this paper since it (1) uses a statistically rigorous form of uncertainty quantification (conformal prediction) rather than heuristics, (2) it can be seen as an alternative way to model uncertainty quantification for VLN (over high-level semantics rather than low-level actions), which is related to reviewer **BZM3**’s comment, (3) it is a much simpler pipeline (e.g., no need for RL) than the proposed approach, but also works on robot hardware and is highly sample-efficient. The works from [2-4] are also highly relevant, as they establish a foundation from the robotics community on how to do vision-language navigation; at minimum they should be cited and contextualized, but some of the works (e.g., [4]) should also be considered fair baselines.

**Reviewer Concerns:**

**7XEJ Concerns**
* (1) lack of clarity in the figures: $\rightarrow$ **hard to evaluate**. It is hard to tell the main changes from the manuscript because changes are not highlighted, so I cannot confirm how effectively these were executed.
* (2) lack of comparison to adaptive inference baselines: $\rightarrow$ **addressed (experiments)**
* (3) ablations to study the sensitivity of the method: $\rightarrow$ **addressed (experiments)**
* (4) detailed questions about the method:  $\rightarrow$ **addressed (experiments)**

**BZM3 Concerns**
* (1) alternative forms of uncertainty quantification beyond action entropy: $\rightarrow$ **addressed (experiments)**
* (2) heuristic reward-shaping $\rightarrow$ **addressed (experiments)**
* (3) not fully demonstrating the “plug and play” nature of the method: $\rightarrow$ **addressed**

**a5ZA Concerns**
* (1) paper’s presentation  $\rightarrow$ **partially addressed**. It is hard to tell the main changes from the manuscript because changes are not highlighted, so I cannot confirm how effectively these were executed. However, I do agree that the presentation is still cramped and hard to read.
* (2) if outdoor scenarios were considered in the evaluation:  $\rightarrow$ **addressed (empirically: added eval on outdoor VLN benchmarks and shows that method underperforms)**

**Reviewer Scores:**

* **7XEJ** would have *maintained* a score of 4: marginally below the acceptance threshold.
* **BZM3** would have *increased* to a score of 6: marginally above the acceptance threshold.
* **a5ZA** would have *maintained* a score of 8: accept, good paper (poster).

---

### Decision · Program_Chairs · 2026-01-26

Reject